# An Evaluation of Nanoparticle Distribution in Solution-Derived YBa$_2$Cu$_3$O$_{7-\delta}$ Nanocomposite Thin Films by XPS Depth Profiling in Combination with TEM Analysis

Els Bruneel *, Hannes Rijckaert, Javier Diez Sierra, Klaartje De Buysser and Isabel Van Driessche

Sol-Gel Centre for Research on Inorganic Powders and Thin Films Synthesis (SCRiPTS), Department of Chemistry, Ghent University, Krijgslaan 281-S3, 9000 Ghent, Belgium; hannes.rijckaert@ugent.be (H.R.); javier.diezsierra@ugent.be (J.D.S.); klaartje.debuysser@ugent.be (K.D.B.); isabel.vandriessche@ugent.be (I.V.D.)
* Correspondence: els.bruneel@ugent.be

**Abstract:** This work discusses the development of an analysis routine for evaluating the nanoparticle distribution in nanocomposite thin films. YBa$_2$Cu$_3$O$_{7-\delta}$ (YBCO) nanocomposite films were synthesized via a chemical solution deposition approach starting from colloidal YBCO solutions with preformed nanoparticles. The distribution of the nanoparticles and interlayer diffusion are evaluated with X-ray photoelectron spectroscopy (XPS) depth profiling and compared with cross-sectional transmission electron microscopy (TEM) images. It is shown that the combination of both techniques deliver valuable information on the film properties as nanoparticle distribution, film thickness and interlayer diffusion.

**Keywords:** XPS; TEM; depth profiling; distribution; nanoparticles; chemical solution deposition

## 1. Introduction

Alongside the incorporation of self-assembled or preformed metal oxide nanoparticles of less than 10 nm in diameter as artificial pinning centers came the need for evaluation of the nanoparticle distribution in the superconducting $RE$Ba$_2$Cu$_3$O$_{7-\delta}$ ($RE$BCO, $RE$ = rare earth) nanocomposite films [1–3]. In this work, the cost-effective chemical solution deposition technique starting from colloidal YBa$_2$Cu$_3$O$_{7-\delta}$ (YBCO) solutions was used to fabricate YBCO nanocomposite films. In this so-called ex situ approach, colloidally stable nanoparticles were synthesized and incorporated in a YBCO precursor solution, providing more control over the final nanoparticles properties (e.g., size and distribution) in the YBCO matrix [4–6]. To date, some attempts have been made in the fabrication of superconducting nanocomposite films using preformed nanoparticles as artificial pinning centers in the trifluoracetic-based YBCO method [4,5,7,8]. The success of this method has been limited because the nanoparticles are either pushed to the YBCO surface or accumulated at the substrate interface. The latter hampers the epitaxial growth of YBCO, leading to poor superconducting properties [5]. It is important that the nanoparticles are homogeneously distributed throughout the superconducting nanocomposite film to maximize the ability to pin the vortices [3].

Nanoparticle distribution can be studied via cross-sectional view transmission electron microscopy (TEM), as it provides topographical, morphological, compositional and crystalline information in 2D on small sample areas, but TEM lamella requires a tedious sample preparation and is also time consuming [5,9–11]. The same remarks hold for the 3D electron tomography, which was used to study the location of nanoparticles in Inconel 718 superalloys [12] or silicon nanoparticles embedded in SiO$_2$ using plasmon tomography [13]. The small sample size and extensive sample preparation were surmounted in the study of metal nanoparticles using secondary-ion mass spectroscopy (SIMS) depth profiling by Peeters et al. [14]. Priebe et al. [15] even extended towards 3D information

using time-of-flight SIMS on a custom-made metal-metal composite and Rijckaert et al. [16] also used SIMS to determine the impact of the stabilization ligands on the distribution of metal ions in the YBCO matrix after the deposition and pyrolysis of the precursor. SIMS has the advantage of a high element sensitivity, which is very useful for dopant profiles in semiconductors, and depth resolution at the interfaces of multilayers [17]. However, a disadvantage is the high dependence of the yield on the matrix composition and structure [18], therefore the range of combinations between the nanoparticles and matrices is restricted.

In this work, the nanoparticle distribution is evaluated via X-ray photoelectron spectroscopy (XPS). XPS is a surface-sensitive technique that has been used in the past to study nanoparticles, i.e., to determine the nature of their surface [19], study interactions with their direct surroundings as ionic liquids [20], or to determine whether they are either encapsulated by a shell or located on the external surface of samples [21,22]. If surface analysis is combined with intermittent sputtering steps, a depth profile can be obtained. This technique has been used extensively for studying thin films [18,22,23]. In previous research, we evaluated the buffer capacity of thin films [24] and surface cleaning methods [25]. Gilbert et al. [26] used XPS combined with sputtering steps for element distribution evaluation rendering information on the distribution of Li ions in layered block polymer parallel films. In addition to being applicable to a wide range of elements or combinations of elements and/or materials, including metals, ceramics and polymers, XPS provides quantitative and chemical bonding information, although one needs to be aware of preferential sputtering and reduction by X-rays. Previous work already demonstrated the use of XPS depth profiling in the analysis of $ZrO_2$ added YBCO [5] and $HfO_2$-added $GdBa_2Cu_3O_{7-\delta}$ [27] nanocomposite films. Recently, Santoni et al. [28] showed that the Zr/Y ratio is a reliable parameter to evaluate zirconium distribution in YBCO matrix.

In this work, the method is applied to study the distribution of $SrTiO_3$ and $BaZrO_3$ nanoparticles in pyrolyzed and crystallized YBCO nanocomposite films. After proper calibration, XPS depth profiling can be used as a mean for thickness measurements [29]. As the sputter rate of an XPS apparatus can vary substantially and even the sputter rate in one spectrometer varies over time, it is necessary to obtain the sputter rate of studied material relative to a known standard. Hence, the sputter rate of nanoparticles added YBCO before and after the crystallization process is determined relative to $Ta_2O_5$. TEM analysis was introduced in this work to validate the XPS results.

## 2. Materials and Methods

### 2.1. Chemical Solution Deposition of the YBCO Nanocomposite Film

The preparation of the colloidal YBCO solutions employed in this work followed the procedure established by Rijckaert et al. [9]. In summary, the YBCO precursor solution was prepared by dissolving $Y(C_3H_5O_2)_3$, $Ba(CF_3CO_2)_2$, and $Cu(C_3H_5O_2)_2$ in methanol with a Y:Ba:Cu stoichiometric ratio of 1:2:3. $SrTiO_3$ and $BaZrO_3$ nanoparticles were synthesized via the solvothermal microwave-assisted method, as described in the work of Diez-Sierra et al. [10]. $SrTiO_3$ nanoparticles have a diameter size of $3.7 \pm 0.5$ nm, while $BaZrO_3$ nanoparticles have a diameter size of $3.2 \pm 0.5$ nm, both measured via transmission electron microscopy. $ZrO_2$ and $HfO_2$ nanoparticles, used for the determination of sputter rate, are synthesized according to De Keukeleere et al. [30]. These nanoparticles were stabilized in a polar solvent (e.g., methanol) and added to the YBCO precursor solution. Colloidal YBCO solutions containing 15 mol.% $SrTiO_3$ or 10 mol.% $BaZrO_3$ nanoparticles were spin coated on (100) $LaAlO_3$ (CrysTec GmbH, Berlin, Germany) single-crystal substrates at 2000 rpm for 1 min and pyrolyzed by heating to 400 °C with a heating rate of 1–5 K/min in a flowing wet $O_2$ atmosphere (1 L min$^{-1}$) to remove the organic components. Subsequently, the as-pyrolyzed films were crystallized at 800 °C in a flowing wet 200 ppm $O_2$ in $N_2$ atmosphere (2 L min$^{-1}$) (more experimental details are described in the work of Rijckaert et al. [31]).

### 2.2. XPS Depth Profiling

XPS depth profiling was carried out with an S-probe XPS spectrometer (Surface Science Instruments, VG, Mountain view, CA, USA) with a monochromatic Al source (1486 eV). An electron gun was set at 3 eV. The argon pressure was controlled with a thermo valve. An adhesive Cu tape acts as a bridge between the stage and the sample and improves the electrical contact. An ISO Technical Report [32] summarizes the various means to determine the sputtered depth and was used as a guideline for the sputtering steps. In this work, a $Ta_2O_5$ reference sample of 1000 nm was used. In the sputter process, an area of $3 \times 5$ mm$^2$ was sputtered using an Ar$^+$ ion gun (4 keV). In between sputtering steps, a spot of $250 \times 1000$ μm$^2$ was analyzed with a pass energy of 90 eV, step 0.1 eV, to ensure a high signal-to-noise ratio and a step of 0.1 eV. The total sputter time and number of cycles were chosen based on the sputter rate, thickness of the thin film and the expected properties of the film. The composition of the film at different depths in the film was estimated using Shirley background subtraction and sensitivity factors as available in the software package CasaXPS (Casa Software Ltd., Teignmouth, UK). Regions for C 1s, Y 3d, O 1s, La 3d5/2 and Ti 2p3/2 were analyzed for $SrTiO_3$ added YBCO nanocomposite films. Peaks of Cu, Al, Ba and Sr were only analyzed in a survey measurement on the surface of the sample (survey spectra are available in the Supplementary Materials). For monitoring the distribution in $BaZrO_3$-added or $ZrO_2$-added YBCO nanocomposite films, the Zr 3p peak was chosen for analysis, thus avoiding the overlap between the Ba 4p3/2 and Zr 3d peaks. For the determination of the sputter rate of $HfO_2$-added YBCO nanocomposite films, the distribution was studied using the Hf 4d5/2 peak.

### 2.3. Microstructural Characterization

Cross-sectional lamella were obtained by an in situ lift-out procedure on an FEI (Hillsboro, OR, USA) Nova 600 Nanolab Dual Beam Focused Ion Beam (FIB) Scanning Electron Microscope (SEM). High-resolution and scanning transmission electron microscopy (HRTEM and STEM) images were taken using a Cs-corrected JEOL (Tokyo, Japan) JEM-2200FS operated at 200 kV with a high-angle annular dark field (HAADF) or bright-field (BF) detector. The composition of the nanocomposite film was determined via energy dispersive X-ray (EDX) spectroscopy in HAADF-STEM mode.

## 3. Results and Discussion

### 3.1. Depth Profiling

Figure 1 shows the depth profile of a pyrolyzed (400 °C) YBCO film with 15 mol.% $SrTiO_3$ nanoparticles on an $LaAlO_3$ single-crystal substrate. Based on the peak areas observed in the regions of C 1s, La 3d5/2, Ti 2p3/2 and Y 3d, the depth profile with relative atom compositions was generated. For clarity of the figure, the data for O 1s have been omitted.

The profile obtained in sputter depth profiling is determined by the real concentration profile, as the interface effects of surface roughness and diffusion are at play. Additionally, there is a broadening of experimental depth profiling due to sputter-induced atomic mixing, the information depth of photoelectrons and potential preferential sputtering. Mixing and information depth change with the concentration, and thus change gradually upon the approach of the interface. In the simplest model, there is a Gaussian decay and the actual interface between the film and substrate should be defined at the inflection of the Gaussian decay of the element concentration. Another commonly used approach to determine the interface is to consider the time until one of the components of the thin film decreases until reaching 50% of the value in the bulk of the film [29]. Yet, in these multicomponent systems, subject to diffusion and suffering from varying carbon concentrations both influencing observed relative concentrations, this is a less straightforward choice. Thus, in Figure 2, the concentration of La at a given depth z in the sample, La(z), is compared to its maximum observed concentration in the $LaAlO_3$ substrate, La(0), which is reached after 30,000 s of sputtering. The derivative of this normalized concentration profile is used to determine the

interface between the YBCO film and the substrate, and is thus estimated to be reached after 24,000 s of sputtering. The profile of the decay in the yttrium concentration confirms this estimation.

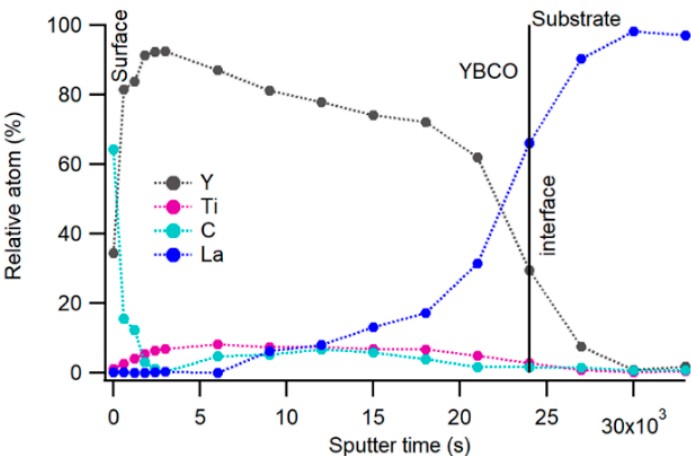

**Figure 1.** Depth profile of pyrolyzed 15 mol.% SrTiO$_3$-added YBCO nanocomposite film on an LaAlO$_3$ substrate, depth profile with oxygen in the Supplementary Materials.

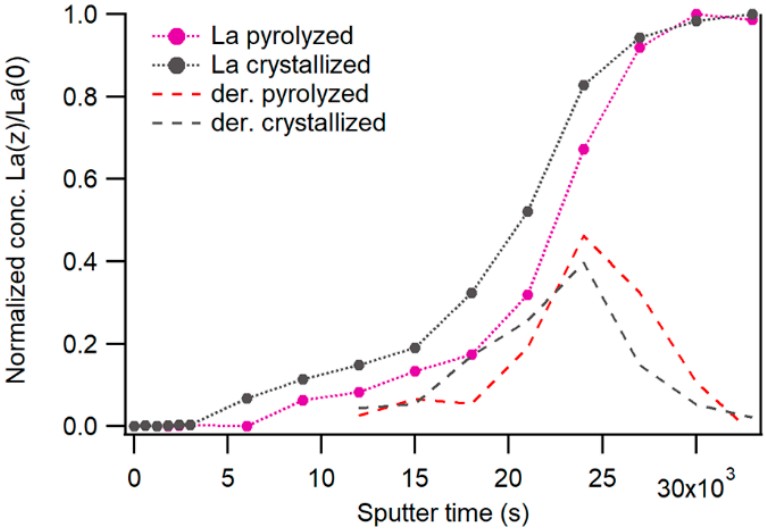

**Figure 2.** Normalized concentration of lanthanum throughout the spectrum before and after crystallization and the first derivatives (dashed lines).

The first signs of lanthanum appears in the spectrum after 9000 s of sputtering (Figure 1) or approximately at 300 nm depth starting from the surface. The yttrium concentration gives the impression that it reduces with sputtering, however this observation is vastly determined by the lanthanum diffusion in the thin film.

The spectrum at the surface before sputtering shows a high concentration of carbon due to the surface contamination. This carbon concentration is rapidly decreasing, but does not completely disappear. One could argue that this can be explained by the redeposition of carbon as the result of desorption from the walls of the vacuum chamber and potential back-streaming contamination from pumps. However, C 1s peak area is measured immediately after each sputter step and takes only 3 min. The redeposition of a detectable carbon signal in the equipment takes an extended redeposition and measuring time. It is apparent that,

within the pyrolysis step, the carbon was not completely removed from the $SrTiO_3$-added YBCO nanocomposite film.

### 3.2. Distribution of SrTiO₃ Nanoparticles in the Pyrolyzed YBCO Film

Atomic concentrations of the nanoparticles tend to be low, compared to other components. In order to evaluate the distribution of the nanoparticles within the film, it is convenient to make a direct comparison (Figure 3) of the yttrium versus titanium signals: $\%Ti/(\%Ti + \%Y)$. Due to carbon contamination, the signals taken at the surface of the sample are far too small to be reliable and were excluded from the figures. On top of the sample, there is a moderate change in the composition of the titanium versus yttrium, but then the distribution is homogeneous over a depth of several hundreds of nanometers, about 7/8 of the total thickness of the film. It is confirmed via the HAADF-STEM analysis (Figure 4A) of the pyrolyzed 15 mol.% $SrTiO_3$-added YBCO nanocomposite film, where no agglomerations of $SrTiO_3$ nanoparticles could be observed.

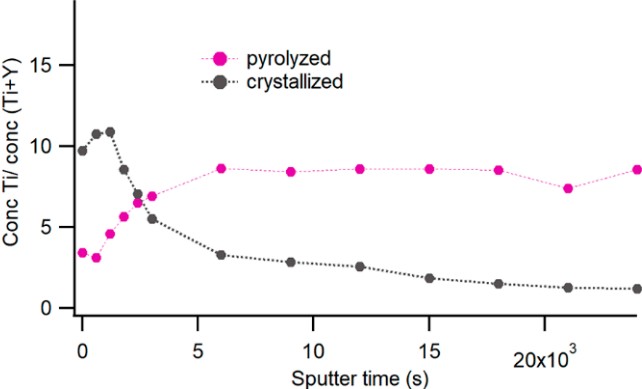

**Figure 3.** Atom percent of titanium versus yttrium along the depth of a pyrolyzed and crystallized $SrTiO_3$-added YBCO nanocomposite film.

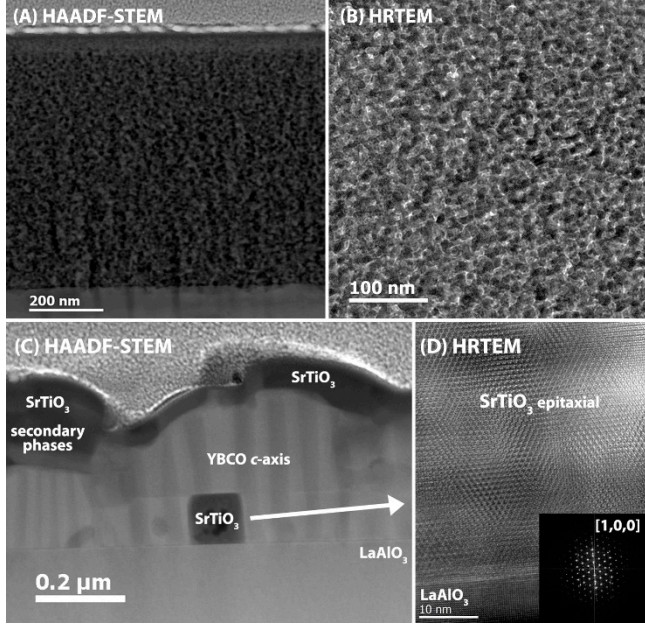

**Figure 4.** Cross-sectional HAADF-STEM images of (**A**) pyrolyzed and (**C**) crystallized $SrTiO_3$-added YBCO nanocomposite films with its HRTEM images (**B**) pyrolyzed and (**D**) crystallized at high magnifications. The inset shows fast Fourier transform (FFT) patterns of epitaxial $SrTiO_3$ phase growing on an $LaAlO_3$ substrate.

### 3.3. The Distribution of SrTiO₃ Nanoparticles in Crystallized YBCO Films

A thermal treatment at 800 °C is necessary to form a YBCO phase. During this process, the film densifies, resulting in a denser and thus thinner film and decreases in sputter rate to a comparable extent. As a result, the crystallized YBCO films (380 nm as determined by HAADF-STEM, Figure 4C) take about the same time to sputter as the thicker pyrolyzed YBCO film (700 nm, Figure 4A). The evaluation of the depth profile and nanoparticle distribution were carried out, as described above. The carbon signal almost completely disappeared from the spectra. The depth profile analysis of a crystallized SrTiO₃-added YBCO nanocomposite film is shown in Figure 5. The top of the derivative of the normalized lanthanum concentration (Figure 2) is again found at 24,000 s of sputtering, however, in this case of a crystallized film, the peak of the derivative is not symmetrical and broader, allocated to diffusion. A first sign of the presence of lanthanum is observed after 6000 s or approximately 100 nm from the surface. We can conclude that the lanthanum- free part of the film is reduced to only the upper 25% of the film.

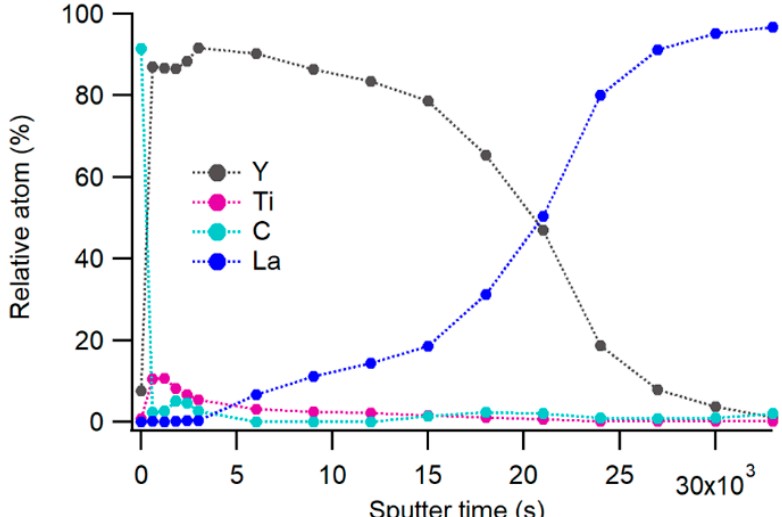

**Figure 5.** Depth profile of crystallized 10 mol.% SrTiO₃-added YBCO nanocomposite film on an LaAlO₃ substrate, depth profile with oxygen in the Supplementary Materials.

During the crystallization process, most of SrTiO₃ nanoparticles migrate towards the YBCO surface. In Figure 3, we can observe an increase in the Ti/Y + Ti ratio at the YBCO surface. This ratio gradually decreases in the bulk of the YBCO film, until only a few percentages of titanium are registered at the interface reached after 24,000 s. To study and understand this migration effect, the nanoparticle concentration was substantially increased to 25 mol.% in the YBCO film. This augments the migration process and is observed on a continuous decrease in the concentration of the titanium (Figure 6) and confirms that the addition of SrTiO₃ nanoparticles is detrimental for the homogeneous distribution. In accordance with HAADF-STEM (Figure 4C), the agglomeration of SrTiO₃ nanoparticles at the YBCO surface and on the LaAlO₃ substrate can be observed. It indicates that SrTiO₃ nanoparticles agglomerated during the crystallization step everywhere, but often at the YBCO surface. When the SrTiO₃ nanoparticles agglomerated at the LaAlO₃ substrate, it grows epitaxial, as confirmed via HRTEM and its fast Fourier transform patterns (Figure 4D). SrTiO₃-added YBCO nanocomposite films exhibit the (00ℓ) YBCO structure and yield fairly good superconducting properties of self-field critical current densities of 3.5–4 MA/cm² at 77 K. As SrTiO₃ nanoparticles are agglomerated and pushed to the YBCO surface, no pinning effect is observable when the magnetic field is increased, which means that SrTiO₃ nanoparticles do not act as pinning centers (more details are available in the Supplementary Materials).

As SrTiO$_3$ nanoparticles are mostly pushed to the YBCO surface, leading to less pinning properties, SrTiO$_3$ nanoparticles are replaced with BaZrO$_3$ nanoparticles. It has been shown in previous work [10] that the introduction of BaZrO$_3$ nanoparticles improved the homogeneity of the nanoparticle distribution in a YBCO matrix. It is also confirmed in Figure 7A, in which the relative percentage of zirconium compared to yttrium %Zr/(%Zr + %Y) is shown in a sample with a 10 mol.% BaZrO$_3$-added YBCO film. It is also confirmed via the cross-sectional BF-STEM image (Figure 7B) that BaZrO$_3$ nanoparticles are homogenously distributed in the YBCO matrix. This homogeneity of BaZrO$_3$ nanoparticles also resulted in an increase in pinning force densities [10]. In comparison to BaZrO$_3$-added YBCO nanocomposite films, the difference between the nanoparticle diameter size of BaZrO$_3$ (3.2 ± 0.4 nm) and SrTiO$_3$ (3.7 ± 0.5 nm) nanoparticles is negligible and that SrTiO$_3$ particles are pushed to the YBCO surface during growth, while BaZrO$_3$ nanoparticles are incorporated in the YBCO matrix after the thermal process. Therefore, this can probably be related to the smaller lattice mismatch between SrTiO$_3$ (3.91 Å) and LaAlO$_3$ (3.81 Å), compared to the lattice mismatch between BaZrO$_3$ (4.25 Å) and LaAlO$_3$ (3.81 Å). Another possibility is explained in the work of Cayado et al. [4], where a difference in CeO$_2$ nanoparticles (2 nm vs. 6 nm) appeared to have an effect on the nanoparticle behavior during the YBCO growth. The small (2 nm) nanoparticles are pushed to the YBCO surface, whereas the large (6 nm) nanoparticles are incorporated in the YBCO matrix, leading to the formation of BaCeO$_3$ particles (4.38 Å). The behavior of nanoparticles during the YBCO growth, i.e., the interaction of nanoparticles at the growth interface, is influenced by several critical parameters, such as the nanoparticle size and the growth rate. Another method is to increase the YBCO film growth rate through adjusting the process parameters, such as the crystallization temperature, heating rate and water pressure.

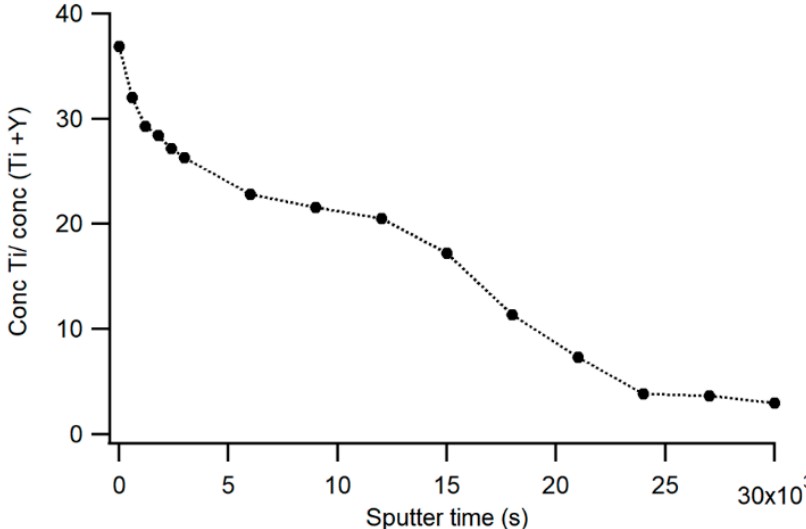

**Figure 6.** The relative concentration of Ti versus Y in a 25 mol.% STO in a YBCO thin film.

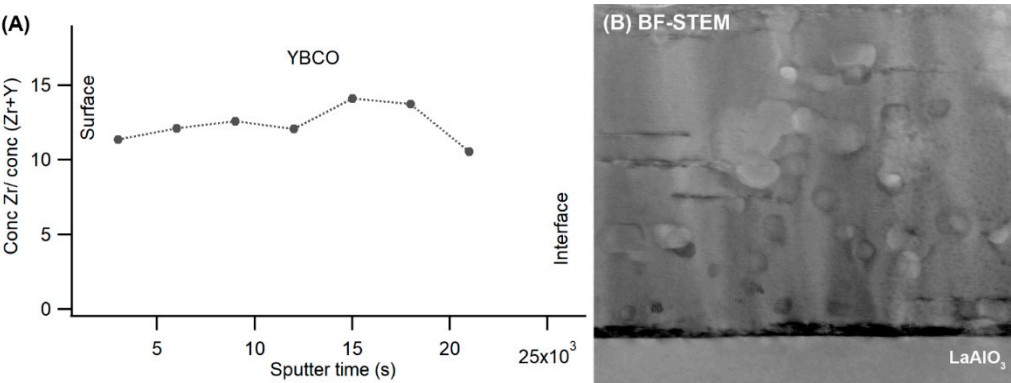

**Figure 7.** (**A**) The relative concentration of zirconium versus yttrium and (**B**) cross-sectional BF-STEM image of $BaZrO_3$-added YBCO nanocomposite films.

### 3.4. The Estimation of the Relative Sputter Rate

The sputter rates of metal oxides can vary over a factor of two [29] and the sputter rate of the spectrometer can vary over a factor of eight over a broad time span, including technical difficulties. In order to plan the lengthy measurements, it was extremely useful to have a knowledge of the relative sputter rate of the pyrolyzed and crystallized nanocomposite films compared to a $Ta_2O_5$ reference sample, used for calibration. With this knowledge, the thickness of unknown films can be estimated.

The thickness of several YBCO nanocomposite films, containing either $SrTiO_3$, $BaZrO_3$, $HfO_2$, or $ZrO_2$ nanoparticles, was measured using cross-sectional TEM images. To calculate the sputter rate, this thickness was compared with the time needed to reach the $YBCO/LaAlO_3$ interface in the XPS experiment, based on the evolution of the concentration profile of yttrium. Depth profiles of 7.5 mol.% $HfO_2$- and 10 mol.% $ZrO_2$-added YBCO nanocomposite films are shown in the Supplementary Materials. Each sputter rate was then compared to the sputter rate of the $Ta_2O_5$ reference sample, which was measured on a regular basis to compensate for periodical changes.

For crystalized YBCO nanocomposites, a relative sputter rate between 0.035 nm/s and 0.11 nm/s was calculated. When comparing these values to the sputter rate of $Ta_2O_5$ at that moment in time, for the equipment used, the relative sputter rate was found to be 0.79 nm YBCO/ nm $Ta_2O_5$ with a standard deviation of 0.15 nm/nm $Ta_2O_5$. (Figure 8). For pyrolyzed films, the sputter rate showed a larger variation, as did the synthesis procedure. An average value of $2.08 \pm 0.63$ nm YBCO/nm $Ta_2O_5$ for samples pyrolyzed between 300 °C and 400 °C was determined. No relation between the type of nanoparticle composition and the sputter rate, and only a weak correlation between nanoparticle concentration and sputter rate could be observed.

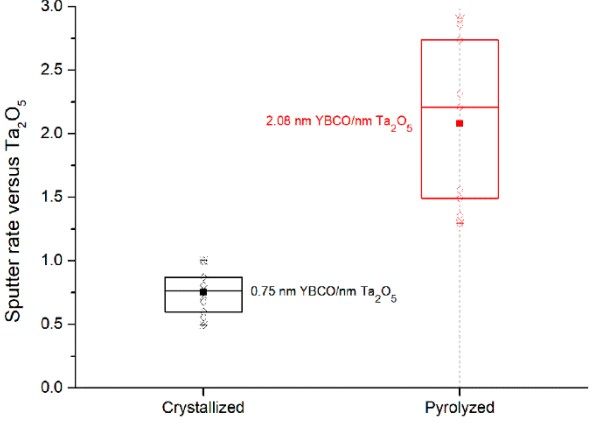

**Figure 8.** YBCO sputter rate versus $Ta_2O_5$ for pyrolyzed and crystallized samples.

## 4. Conclusions

Cross-sectional view electron microscopy of the nanocomposite films can present the possibility to measure the correct film thickness and also the information on the homogeneity of nanoparticles and its size in YBCO films after thermal process (pyrolysis and crystallization). However, TEM analysis offers a small sample area (10 $\mu m^2$) measurement and the FIB preparation to create TEM lamella is very tricky and time consuming.

In this work, we demonstrated that XPS depth profiling is a quantitative analysis to measure an area of 0.25 $mm^2$. This method presents a reliable delineation of the nanoparticle distribution in the YBCO matrix and that the film thickness can be estimated after calibration. Here, $BaZrO_3$, $ZrO_2$ and $HfO_2$ nanoparticles are embedded into the YBCO matrix due to its higher lattice mismatch, while $SrTiO_3$ nanoparticles with a lower lattice mismatch are pushed to the YBCO surface. This work not only shows that the nanoparticle size should be in the range of the critical nanoparticle size to avoid the pushing or accumulation effect, but also that the lattice mismatch of nanoparticles should be higher to be incorporated in the YBCO matrix. XPS in combination with TEM is a useful method to obtain an in-depth analysis of nanoparticle distribution, regardless of their chemical composition, which showed to be highly complementary in nanocomposite film research.

**Supplementary Materials:** The following supporting information can be downloaded at: https://www.mdpi.com/article/10.3390/cryst12030410/s1, Figure S1: Depth profile of pyrolyzed 15 mol.% $SrTiO_3$-added YBCO nanocomposite film on an $LaAlO_3$ substrate, including the oxygen signal; Figure S2: Depth profile of crystallized 10 mol.% $SrTiO_3$-added YBCO nanocomposite film on an $LaAlO_3$ substrate, including the oxygen signal; Figure S3: Survey of the surface of a crystallized 10 mol.% $ZrO_2$-added YBCO nanocomposite film on an $LaAlO_3$ substrate; Figure S4: Depth profile of crystallized 10 mol.% $ZrO_2$-added YBCO nanocomposite film on an $LaAlO_3$ substrate; Figure S5: Survey of the surface of crystallized 10 mol.% $ZrO_2$-added YBCO nanocomposite film on an $LaAlO_3$ substrate; Figure S6: Depth profile of crystallized 10 mol.% $ZrO_2$-added YBCO nanocomposite film on an $LaAlO_3$ substrate; Figure S7: Survey of the surface of pyrolyzed 7.5 mol.% $HfO_2$-added YBCO nanocomposite film on an $LaAlO_3$ substrate; Figure S8: Depth profile of pyrolyzed 7.5 mol.% $HfO_2$-added YBCO nanocomposite film on an $LaAlO_3$ substrate; Table S1: Critical current density ($J_c$) of 0, 5, 10 and 15 mol.% $SrTiO_3$-added YBCO films on an $LaAlO_3$ substrate, calculated via inductively measurements; Figure S9: Superconducting properties of 5 and 10 mol.% $SrTiO_3$-added YBCO nanocomposite films compared with an pristine YBCO film: (A) Magnetic field dependence of critical current density at 77 K and (B) angular dependence of $J_c$ at 77 K and 1 T.

**Author Contributions:** E.B. wrote the manuscript, and performed and analyzed the XPS measurements; J.D.S. synthesized and stabilized the nanoparticles; H.R. deposited the nanocomposite films and carried out the TEM measurements; I.V.D. and K.D.B. provided supervision. All authors have read and agreed to the published version of the manuscript.

**Funding:** This work was financially supported by the European Union Horizon 2020 Marie Curie Actions under the project SynFoNY (H2020/2016-722071). H.R. acknowledges the support and funding as postdoctoral fellow fundamental research of the Research Foundation—Flanders (FWO) under grant number 1273621N.

**Institutional Review Board Statement:** Not applicable.

**Informed Consent Statement:** Not applicable.

**Data Availability Statement:** The data presented in this study are available on request from the corresponding author.

**Acknowledgments:** The authors thank Nico De Roo for his assistance with the XPS measurements.

**Conflicts of Interest:** The authors declare no conflict of interest.

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
