# Peer review of "An Evaluation of Nanoparticle Distribution in Solution-Derived YBa2Cu3O7−δ Nanocomposite Thin Films by XPS Depth Profiling in Combination with TEM Analysis"

_crystals, doi:10.3390/cryst12030410_

Round 1

Reviewer 1 Report

The manuscript is interesting, the study of the distribution of elements in a superconducting film and, in particular, the "film-substrate" interfaces is an urgent trend in modern materials science. However, there are a number of significant comments that need to be corrected:
1) there is no information about the pass energy when registering core levels and survey spectra. 90 eV for core levels is very large;
2) it is necessary to attach a set of survey spectra to additional materials as well as core-level spectra of YBCO.
3) should be shown in Fig. 1 and 5 additional figures with oxygen or add oxygen data to the figures;
4) you write "For crystallized YBCO nanocomposites a relative sputter rate of 0.75 nm". However, sputter rate is measured in nm/s or nm/min. This information needs to be corrected.
5) there are no element distribution profiles for YBCO nanocomposite films with BaZrO3 added or ZrO2, as well as HfO2 added YBCO. Why? What is the difference in profiles when adding these particles?

Reviewer 2 Report

I read with interest the paper titled "Evaluation of nanoparticle distribution in solution-derived YBa2Cu3O7-δ nanocomposite thin films by XPS depth profiling in combination with TEM analysis." The authors report on detailed investigation of CSD-derived YBCO films, in which SrTiO3 or BaZrO3 nanoparticles are distributed. Using XPS they obtain information on nanoparticle distribution, film thickness and interlayer diffusion. 

The manuscript is well organized, written and is of interest for researchers working on solution-derived oxide thin films. I agree with its publication after minor comments listed below are addressed. 

What is the driving force for agglomeration of SrTiO3 on the top of the film? Why BaZrO3 does not agglomerate on the surface? Please comment in the paper. 

Is particle size of and its distribution of BZO and STO nanoparticles the same? Please add the values and comment the differences. 

Figure 7: the BF-STEM image is cut-out of the view. Also, it is not commented in the text. Please update. 

Reviewer 3 Report

The results of study presented in the manuscript “Evaluation of nanoparticle distribution in solution-derived YBa2Cu3O7-δ nanocomposite thin films by XPS depth profiling in combination with TEM analysis.” by  Els Bruneel et al. connected with the development of an analysis routine for evaluating the film thickness, interlayer diffusion and nanoparticle distribution in nanocomposite YBa2Cu3O7-δ thin films using X-ray photoelectron spectroscopy (XPS) depth profiling and cross-sectional  transmission electron microscopy (TEM) images. The manuscript is contained new interesting experimental results and analytical approach aimed on the solution of the assigned tasks. The manuscript can be published after some corrections and additions. But the given data are not properly discussed in connection with quantitative characteristics of the nanoparticles distribution in the film, for example, via its thickness or volume as well as no information is given about particles sizes present in the films or the degree of their agglomeration.

Comment 1

In the conclusion part is given the following statement “In this work, we have demonstrated that XPS depth profiling is a quantitative analysis to measure an area of 0.25 mm2. This method gives a reliable delineation of the nanoparticle distribution in YBCO matrix… It has universal applicability to obtain in-depth analysis of nanoparticle distribution, regardless of their chemical composition.

In fact the authors estimated how varied the composition of 0.25 mm2  area of the film  in depth depending on time of spattering, as well as variation of the ratio of atom percent of titanium versus titanium plus yttrium depending on time of sputtering of a pyrolyzed and crystallized films. It is well know that XPS depth profiling can give this information about any material. If possible, the obtained data should be discussed from the point of view of variation in particle sizes and related to the quantitative characteristics of their distribution in the volume of the films.        

Comment 2

It will be interesting to have information about superconducting characteristics of the films under the study.

Comment 3

Graphs shown in Figure 2 need additional discussion. More detailed explanation should be given for the difference between the data for pyrolyzed and crystallized SrTiO3 added YBCO nanocomposite films.

Comment 4

In the text of the manuscript after the reference to figure 1 follows the reference to figure 4a (page 3, lines 123-128)  and references to Figures 2 and 3  one can see on page 3, line 140 and page 4, line 171. Please, correct the sequence of references to the figures.    

Comment 5

Page 2 line 49

“in semiconductors, and depth resolution at the interfaces of multilayers.[17]”

The dot in the sentence should be after the reference to the literary source.

Page 8, line 267 (Conclusion)

“… in nanocomposite film research..”

Extra dot is at the end of the sentence.

Comment 6

The borders of Figure 7 extend beyond the margins of the page. Please, correct.

Round 2

Reviewer 1 Report

All my wishes are taken into account. The updated manuscript may be published.